# OpenReview forum: "Agent-to-Agent Theory of Mind: Testing Interlocutor Awareness among Large Language Models"
_colmweb.org/COLM/2025/Workshop/Social_Sim — Social Sim'25_

### Official Review · Reviewer_B33j · 2025-07-17
**Agent-to-Agent Theory of Mind: Testing Interlocutor Awareness among Large Language Models**

**Rating:** 7
**Overall Assessment:** 4
**Confidence:** 3

**Review:**

This paper tackles the problem of interlocutor awareness in LLMs with three diagnostic approaches and three case studies, showing strong self‑recognition and practical impacts on co‑operation, reward hacking, and jailbreaks.
Its strengths lie in topical relevance, breadth across five major model families, and transparency (the appendix lists every prompt and dataset), making the work easy to follow. However, the empirical scale is modest, so several cross‑family gaps may be statistical noise. Further limitations include reliance on a single prompt template (risking lexical artefacts) and identity granularity restricted to family‑level (ignoring, e.g., GPT‑4o vs GPT‑4o‑mini). An earlier, richer related‑work discussion would better situate the contribution. Overall, the study is original and significant for multi‑agent LLM research, but its conclusions would be stronger with larger samples, and finer‑grained analyses.

**Comments Suggestions And Typos:**

- If budget allows, increase the number of prompts per dataset (or pool multiple sets) to tighten confidence intervals. The present $\approx100$ examples leave large statistical uncertainty.
- Introduce paraphrased adversarially re‑styled templates to show that identification is not driven by superficial lexical artefacts.
- Consider a fine‑grained experiment that discriminates between versions of the same family (e.g. GPT‑4o vs GPT‑4o‑mini). This would test whether the approach generalises beyond coarse family labels.
- Typo: Pattenrs (Fig. 1)

**Paper Summary:**

The paper focuses on the interlocutor awareness of the LLMs. Looking into their ability to detect what family of language models they are encountering. Using cues from reasoning patterns, linguistic style, and alignment preferences, the authors show strong self‑recognition and moderate cross‑family detection across GPT, Claude, DeepSeek, Llama 3, and Mistral. They perform three short case studies and then demonstrate the impact: identity knowledge boosts cooperative math solving, enables reward‑hacking against known judges, and slightly raises jailbreak success.

**Relevance:**

4

**Summary Of Strengths:**

1. Interlocutor awareness is increasingly critical as multi‑agent LLM workflows gain traction, making this investigation well‑aligned with current research priorities.
2. Beyond core benchmarks, the paper analyses three high‑impact scenarios, clearly demonstrating why interlocutor awareness matters in practice.
3. The manuscript reads smoothly, and the appendix lists all prompts, datasets, providing a thorough and reproducible account of the work.

**Summary Of Weaknesses:**

1. Prior work on model attribution and situational awareness is introduced after the main results and only briefly, making it hard to situate the contribution and assess novelty while reading. Given the topic is new, I would recommend bringing the related works earlier on.
2. Experiments use only around 100 prompts per dataset and provide few implementation specifics. This is problematic because performance can vary widely across LLM checkpoints and sampling settings.

---

### Official Review · Reviewer_QFWY · 2025-07-19

**Rating:** 5
**Overall Assessment:** 3
**Confidence:** 3

**Review:**

Strengths:
- The work proposes and defines "interlocutor awareness" in LLMs
- The empirical evaluation is very comprehensive. They focus on three different dimensions (reasoning, style, preferences) across many different types of benchmarks
- They demonstrate both beneficial (improved cooperation) and adverse (reward-hacking, jailbreak vulnerabilities) effects, highlighting real-world implications.

Weaknesses/Suggestions for improvement:
- The definition of identity is very coarse. They restrict "identity" to model family, omitting other types of granularity such as architecture size, fine-tuning, persona nuances, etc. which may understate richer awareness capabilities.
- They rely on a single core prompt template per condition which may introduce unintended biases; varying prompt structures could affect outcomes.
- Each evaluation only uses 100 randomly sampled examples per dataset.

**Comments Suggestions And Typos:**

See weaknesses above.

**Paper Summary:**

This paper introduces interlocutor awareness, defined as a model's ability to infer and adapt to the identity of its conversational partner. The work focuses on two key questions: (1) Can LLMs identify other LLMs based only on their responses? (2) How does knowing an interlocutor's identity affect behavior in cooperative and competitive scenarios? They design a single-turn and multi-turn identification framework where an identifier model labels a target model's family based on its output. They empirically evaluate five closed-source and six open-source model families on diverse benchmarks.

**Relevance:**

4

**Summary Of Strengths:**

Strengths:
- The work proposes and defines "interlocutor awareness" in LLMs
- The empirical evaluation is very comprehensive. They focus on three different dimensions (reasoning, style, preferences) across many different types of benchmarks
- They demonstrate both beneficial (improved cooperation) and adverse (reward-hacking, jailbreak vulnerabilities) effects, highlighting real-world implications.

**Summary Of Weaknesses:**

Weaknesses/Suggestions for improvement:
- The definition of identity is very coarse. They restrict "identity" to model family, omitting other types of granularity such as architecture size, fine-tuning, persona nuances, etc. which may understate richer awareness capabilities.
- They rely on a single core prompt template per condition which may introduce unintended biases; varying prompt structures could affect outcomes.
- Each evaluation only uses 100 randomly sampled examples per dataset.

---

### Meta-Review · Area_Chair_rXBb · 2025-07-21

**Recommendation:** Accept

**Metareview:**

--